# From Knowledge to Wisdom: Looking beyond the Knowledge Hierarchy

**Constantin Bratianu** [1,*,†] and **Ruxandra Bejinaru** [2,†]

1 UNESCO Department for Business Administration, Bucharest University of Economic Studies, Piata Romana 6, 010731 Bucharest, Romania
2 Department of Management, Business Administration and Tourism, "Stefan cel Mare" University of Suceava, Str. Universitatii 13, 720229 Suceava, Romania; ruxandrab@usm.ro
* Correspondence: constantin.bratianu@gmail.com
† These authors contributed equally to this work.

**Abstract:** Although there is a long history of searching for the road from knowledge to wisdom, there is no final and clear result. In fact, there are multiple ways of starting from knowledge and reaching wisdom due to the complexity of the semantic domains of both concepts. In addition, there are different perspectives on interpreting these conceptual maps, ranging from philosophy to psychology or management. We are interested in understanding the connecting ideas between knowledge and wisdom from the management perspective, where decision making is the key driving force for transforming knowledge into efficient actions for creating value for customers through products and services. The well-known knowledge pyramid or wisdom pyramid is a good metaphor to start with in understanding the basic concepts of data, information, knowledge, and wisdom (DIKW) and their transformations. We analyze different interpretations of these four basic concepts and focus on the transition from knowledge to wisdom, looking beyond the DIKW pyramid. Additionally, to get a larger view of the multiple connections between knowledge and wisdom, we perform a bibliometric analysis using VOSviewer as a specialized software tool. The contribution of the present paper comes from this enlarged framework of searching for links between knowledge and wisdom and analyzing their relevance to business management. The results are relevant to anyone who would like to understand how to manage efficiently knowledge in their organizations. We explain the semantic differences in interpreting the concepts of "information" and "knowledge" in philosophy, information science, and knowledge management, which can be useful both in theory and in practice.

**Keywords:** data; information; knowledge; knowledge pyramid; wisdom; bibliometric analysis

## 1. Introduction

The DIKW (data, information, knowledge, wisdom) hierarchy, also known as the knowledge hierarchy or wisdom hierarchy, is considered by many authors the best framework for defining and understanding the fundamental concepts of data, information, knowledge, and wisdom [1–4]. The credit for this conceptual model is given to Ackoff [5], although the key idea goes back to T.S. Eliot, who wrote in 1934 in "The Rock" [4]:

Where is the Life we have lost in living?
Where is the wisdom we have lost in knowledge?
Where is the knowledge we have lost in the information?

If T.S. Eliot reveals how Life, wisdom, and knowledge disintegrate in their components, the DIKW hierarchy shows how by integrating data, we get information; by integrating information, we get knowledge; and by integrating knowledge, we get wisdom. As Rowley [3] remarks, "The hierarchy is used to contextualize data, information, knowledge, and sometimes wisdom, with respect to one another and to identify and describe the processes involved in the transformation of an entity at a lower level in the hierarchy

(e.g., data) to an entity at a higher level in the hierarchy (e.g., information). The implicit assumption is that data can be used to create information; information can be used to create knowledge, and knowledge can be used to create wisdom" (p. 164).

The DIKW hierarchy is considered a canonical model because it is simple and very intuitive, although it does not explain what the mechanisms and the driving forces which produce each transformation from one given level to the next level of complexity are. In addition, it creates the false idea that these transformations are unique in producing information, knowledge, and wisdom. Even if some authors introduce some new levels, such as intelligence [2], truth [4], or signals [6], the limitations of this hierarchy remain. It is one of our purposes to show these limitations and how to interpret them in the context of knowledge management and the increasing importance of its spiritual dimension.

The main idea of this pyramid is that information, knowledge, and wisdom result from processing the content of their inferior levels through accumulation, synthesis, and filtering according to some criteria. However, information and knowledge have multiple interpretations coming from the theory of information, management, and cognitive science, which cannot be contained within the DIKW hierarchy [7–12]. Starting from this observation based on a critical review of the literature, the present paper aims to go beyond the knowledge hierarchy and reveal the complexity of the semantic synapses between information, knowledge, and wisdom. We formulate the research question as follows:

> RQ: What are the semantic synapses between information and knowledge, and between knowledge and wisdom, from the management perspective?

To enlarge our critical review of the literature, we perform a bibliometric analysis using VOSviewer [13,14] as a specialized software tool.

The present paper is structured as follows. After this short introduction, we perform a critical literature review and then present the methodology used for the VOSviewer analysis. Immediately follows a section with results and discussions, and then the conclusions.

## 2. Literature Review

### 2.1. The DIKW Hierarchy

In Figure 1, we present an illustration of the DIKW hierarchy, with its base on data and having wisdom as its pinnacle.

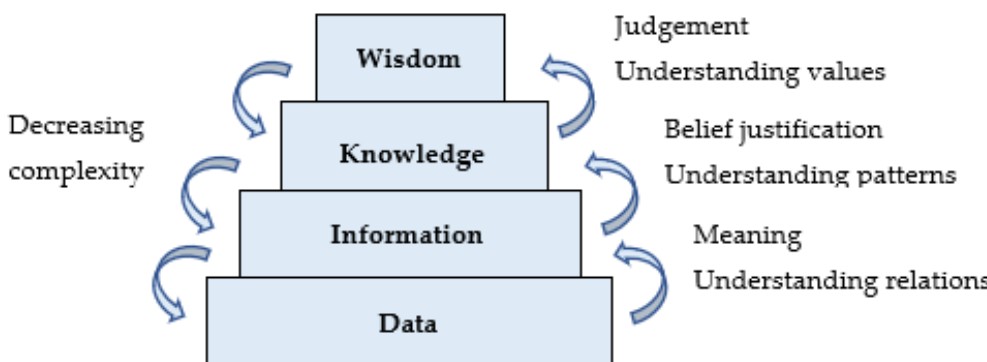

**Figure 1.** The DIKW hierarchy. (Source: Adapted from Rowley [3]).

The DIKW hierarchy is based on a set of four axioms:

(a) Data, information, knowledge, and wisdom represent distinct levels of knowing characterized by different complex content.
(b) The content of each level is obtained by processing the content of the lower level.
(c) Complexity increases in a bottom-up fashion.
(d) Transitions from one level to another one are nonlinear and non-programmable processes.

We would like to stress that Figure 1 represents a simplification of a very complex and mostly unexplored process that is influenced by the education, experience, intelligence,

and culture of each individual. Even if we may distinguish the four levels of knowing complexity, they differ from one individual to another. Additionally, it is important to remark that each level is characterized by a different richness and intensity of knowing, which depend on the experiential and organizational contexts [15–18].

The transition from knowledge to wisdom implies judgment and decision making. According to Baron [19], judgment is an "evaluation of one or more possibilities with respect to a specific set of evidence and goals. In decision making, we can judge whether to take an option or not, or we can judge its desirability relative to other options" (p. 8). Therefore, wisdom is much more than knowledge because it includes judgment.

The knowledge hierarchy remains an important framework for understanding the main levels of knowing and the increasing complexity of knowledge processing from the bottom of the pyramid to its top, although its relevance should not be overestimated. Moreover, it is important to show the dynamics of each transition from data to information, from information to knowledge, and from knowledge to wisdom. These transitions are nonlinear and much more complex than a simple hierarchy may suggest.

*2.2. Data*

Data represents the content of the first level of the hierarchy, and it is composed of letters, numbers, symbols, or different structures of them. According to Davenport and Prusak [20], "Data is a set of discrete, objective facts about events. In an organizational context, data is most usefully described as structured records of transactions" (p. 2). Data has no meaning in itself. It is the raw material for information. For instance, 1, 7, &, T, *, S, =, and F&Y represent data. If we have a table with numbers without any title, we do not know how to interpret them. If we put the title "The age of students in my class", we get a meaning, but if we put another title, "The height of students in my class," we get a different meaning. Computer systems collect, process, and store data and serve as input in any organization's information system. Data has no inherent meaning. "It provides no judgment or interpretation and no sustainable basis for action. While the raw material of decision making may include data, it cannot tell you what to do. Data says nothing about its own importance or irrelevance." [20] (p. 3).

With the new intelligent technologies, data has increased its volume, variety and speed in an exponential way that cannot be handled by conventional management methods and tools. Big Data, Predictive Analytics, and Data Science have emerged as new domains focusing on processing the wealth of data and exploring possible relationships between variables and creating information out of them [21,22].

*2.3. Information*

The concept of information is the critical level of the whole hierarchy of knowledge because of its semantic dimensions coming from mathematics, information science, linguistics, philosophy, and management.

Shannon [7] developed a mathematical theory of communication to provide solutions for the engineering systems of communications in the presence of external perturbations. For simplicity, Shannon considered a generic communication system composed of the following components: (a) a sender who constructs messages to be sent; (b) a transmitter that is a technological device able to transform the message into a structure of electrical or electromagnetic signals; (c) a receiver that is a technological device able to transform the received signals into audio or video, or combined audio and video structures; (d) the end user who interprets the received messages; and (e) a source of external perturbations. The engineering problem formulated by Shannon was to build the capacity to reproduce accurately at one point the message selected at another point. He defined the concept of information as a mathematical entity devoid of any meaning. "Frequently, the messages have meaning; that is, they refer to or are correlated according to some system with certain physical or conceptual entities. These semantic aspects of communication are irrelevant to

the engineering problem" [7] (p. 1). Shannon [7] also defined the concept of information entropy using Boltzamnn's entropy formula:

$$H = -k \, \Sigma \, p(i) \, \log p(i)$$

where k is a constant related to the measuring framework and p(i) represents the probability of occurrence of the event i (i.e., selection of a certain symbol from the database). Therefore, information entropy (H) represents a measure of the entropy of a certain probability distribution. Shannon defined as a measuring unit the bit (abbreviation derived from "binary digit"). Shannonian information and information entropy have remained key concepts in the information science and technology fields due to their power of abstraction and ability to measure the probability distributions of different sets of events or symbols.

Bar-Hillel and Carnap [23] tried to extend the concept of information in linguistics and introduced the idea of semantic information. Practically, they considered a generic language as a finite set of sentences, with each sentence being used with a certain frequency. Thus, their information concept could have an associated meaning within a certain linguistic context. Bar-Hillel and Carnap [23] developed a sophisticated theory that had few chances to be applied due to its simplifying hypotheses. Yet, the basic idea of working with texts and trying to find methods of extracting semantic information from them and constructing qualitative databases remains a challenge for experts in information technology [24].

Floridi [25] recognized that information is still an elusive concept, although its importance to science and technology is crucial today. He focused on information from a philosophical perspective, trying to extend the work done by other researchers in defining semantic information [26,27]. Floridi considered that "semantic information is well-formed, meaningful, and truthful data" [25] (p. 31). From his perspective, the philosophy of information should be concerned with a critical investigation of the conceptual framework of information and its exploration, being aware of the revolution generated by computers and artificial intelligence. He posited that information can be analyzed from three different perspectives: (a) information as reality, focusing on electrical or electromagnetic signals in the same way as Shannon; (b) information about reality, focusing on its semantic dimension; and (c) information for reality, focusing on orders, regulations, and algorithms, such as in knowledge management. Increasing the abstraction level, Floridi [25] provided a general definition of information (GDI) as follows:

"GDI: σ is an instance of semantic information if and only if:

1.  σ consists of n data (d), for n ≥ 1;
2.  the data are well-formed (wfd);
3.  the wfd are meaningful (wfd = δ);
4.  the δ are truthful." (p. 104)

In other words, Floridi [25] considered semantic information as veridical data that is well-structured and meaningful. Thus, the concept of semantic information defined by Floridi [25] comes very close to the concept of knowledge used in knowledge management without having the capacity to replace it. It is interesting to see how Floridi [25–27] used the expression of semantic information promoted by Bar-Hillel and Carnap [23], although he departed from the idea of predefining a finite set of sentences with their associated frequencies or probabilities of being used within a given language. Floridi was interested in finding a general definition for the concept of information rather than computing the entropy of the probability distribution of a certain set of sentences. He attempted to extend the meaning of semantic information toward the concept of knowledge, although there is no clear transition from one level to another one, such as in the hierarchy of knowledge.

In knowledge management, the concept of information is defined with respect to the concepts of data and knowledge. According to the logic of the knowledge hierarchy [1–5], information is a result of processed data. The transition from data to information can be performed by considering a semantic context and a set of correlations between data and the context. "Data becomes information when its creator adds meaning" [20] (p. 4). However,

it is not a simple process of addition but rather an interpretation of data processing with respect to a semantic framework. People process data using their mental models developed through education and personal efforts [28]. As a result of this process, data is invested with meaning and relevance [4,6] with respect to a certain context. From a computational perspective, Davenport and Prusak [20] showed that information emerges from data through the following processes: contextualization, categorization, computation, correction, and condensation.

Although we use the same concept of information in information systems and in knowledge management systems, there are actually three different concepts: Shannonian information, semantic information, and knowledge management information. This semantic paradox generates much confusion in the literature, especially when researchers from the field of information systems discuss issues related to knowledge management systems and make no distinction between the concept of information defined by Shannon [7] and that used in the hierarchy of knowledge [1–5]. The only way of avoiding possible misinterpretations is to understand the emergence of each concept within its specific context and purpose. In addition, it is important to realize that the hierarchy of knowledge represents only an instance of using the concept of information and not the exclusive conceptual framework of defining it, as it frequently happens in the literature.

*2.4. Knowledge*

In the DIKW hierarchy, the concept of knowledge holds the dominant position [4,10,20]. Whenever we discuss the concept of knowledge, we have to distinguish between the philosophical and managerial perspectives. The first and most relevant conceptual framework is given by philosophy [9,10,29,30]. Aristotle [31] considered three categories of knowledge as states of the soul: episteme, techne, and phronesis. Episteme is objective and scientific knowledge. It is a result of our thinking and reflects our need to understand the world we are living in. As Aristotle posited, "what is known scientifically is by necessity" [31] (p. 88). Techne is the craft knowledge that is needed in production. Techne is the knowledge of know-how for producing goods and services. It is similar but not identical to tacit knowledge. Phronesis is more complex because it integrates both knowledge and decision making. In the literature, authors use for phronesis prudence or practical wisdom [10,32,33]. In Aristotle's view, "Prudence is a state grasping the truth, involving reason, concerned with action about things that are good or bad for human being" [31] (p. 89). Thus, prudence or practical wisdom includes values as guidelines in decision making. The concept shows the transition from the level of knowledge to that of wisdom in the DIKW hierarchy.

Both Aristotle and Plato considered that knowledge is a result of thinking and not of senses, a result of reflection and not of impression. In their view, only the mind can reach existence and evidence the truth. "It follows that we cannot know things through the senses alone, since through the senses alone we cannot know that things exist. Therefore, knowledge consists in reflection, not in impression, and perception is not knowledge" [29] (p. 153). This idea was developed further by Descartes [34], who created the theory of dualism of mind and body and attributed the essential role to the mind and rational knowledge. An important contribution to changing that paradigm and recognizing that perception is a part of the learning process was made by Polanyi [35], who defined the tacit dimension of knowing. He posited, "I shall reconsider human knowledge by starting from the fact that we can know more than we can tell" [35] (italics in the original, p. 4). Thus, Polanyi integrated into the learning process experiential knowledge [15] and highlighted the difference between the knowledge that can be expressed using natural or symbolic language and the knowledge that can be expressed only by body language.

Unlike Western philosophy and Descartes' dualism of mind and body, Japanese philosophy is based on the oneness of mind and body and on the idea of integrated knowledge [10]. The Japanese approach is based on three basic ideas: (a) oneness of humanity and nature; (b) oneness of mind and body; and (c) oneness of self and others [36–38]. These ideas reflect the influences of the ancient teachings of Buddhism, Shintoism, Zen, and Con-

fucianism, and they explain the Japanese attraction to the harmony of the whole instead of the efficiency of its parts.

Using the iceberg metaphor and Polanyi's tacit dimension assumption, Nonaka and Takeuchi [10] defined the dyad of explicit knowledge–tacit knowledge and developed the theory of knowledge-creating dynamics. Tacit knowledge is the individual knowledge obtained through experiential learning and processed by the cognitive unconscious part of the brain. It can be expressed directly as body language, and it is transformed into explicit knowledge using a natural or symbolic language [10,12,15,39,40]. "Tacit knowledge is deeply rooted in an individual's action and experience, as well as in the ideals, values, or emotions he or she embraces" [10] (p. 8). This remark is important to keep in mind when considering the DIKW hierarchy to see the limitations of that model of creating knowledge by processing information.

Going beyond this hierarchy, Nonaka and Takeuchi [10] presented their knowledge creation dynamics model composed of four generic operations: socialization, externalization, combination, and internalization (SECI). Socialization is the transfer of tacit knowledge from one individual to another within a social context. It is a frequently used knowledge transfer process in Japanese companies, where collaboration and team spirit are skills learned through education. Thus, knowledge sharing is a part of Japanese life and work. Externalization is an individual process and refers to the transformation of tacit knowledge into explicit knowledge using metaphorical thinking [41–43] and language [38,39]. Combination is the process of knowledge sharing and transforming individual knowledge into collective knowledge through the contributions of other individuals who participate in the process. It is possible for explicit knowledge that is mostly rational. Internalization is the reverse of externalization. It is performed at the individual level and consists of transforming significant explicit knowledge into tacit knowledge. The four processes comprise a cycle that continues in time, generating a knowledge spiral [10,44]. It is interesting to remark that the SECI model has no correlation with the DIKW hierarchy because there is no need to consider information as a raw material for knowledge. The SECI model has been accepted by many researchers and practitioners because it is simple and intuitive. However, the model has some limitations, which should be understood by those who study and apply knowledge management. There is no hard theory behind it, and it can be illustrated with many practical examples. However, the SECI model is more specific to Japanese companies because knowledge sharing is embedded in Japanese education, whereas it is not so clearly adopted in those cultures where there is fierce competition between individuals. In those cultures, the mainstream thinking is that knowledge sharing is a professional vulnerability, and it should be balanced with knowledge hiding [45–47]. At the organizational level, the dynamics of knowledge sharing–knowledge hiding contribute to the changing distribution of knowledge throughout the organization, which leads to the variation of the knowledge entropy [48].

The theory of knowledge fields and knowledge dynamics develops a new perspective for understanding knowledge based on the energy metaphor [49]. The theory asserts that knowledge manifests as a multidimensional field of rational, emotional, and spiritual knowledge. Rational knowledge refers to the knowledge that is a result of rational thinking. It is explicit knowledge. Emotional knowledge is created by our emotions, and it is expressed as body language. Spiritual knowledge refers to the values and principles used in our decision-making process and our behavior. Knowledge from each field can be transformed into knowledge from any other field, generating continuous dynamics. The theory of knowledge fields creates the needed transformation toward wisdom through spiritual knowledge. The process is nonlinear and much more complex than the descriptive model of the DIKW hierarchy.

If we analyze now the dynamics of the DIKW hierarchy, we see that the axiom of knowledge creation by processing information cannot explain the functioning of the SECI model. Additionally, it cannot lead to understanding the theory of knowledge fields. The DIKW hierarchy represents a gross simplification of the complexity of knowledge and its dynamics at both the individual and organizational levels.

*2.5. Wisdom*

Considering the framework given by the DIKW hierarchy, we explain wisdom as a result of processing knowledge [1–5]. However, the concept of wisdom is much more complex than its position in the DIKW hierarchy, even if it is the pinnacle of that hierarchy. Jashapara [4] defined wisdom as "the ability to act critically or practically in a given situation. It is based on ethical judgment related to an individual's belief system" (p. 19). Interpreting this definition, we understand that wisdom is the ability to act with respect to a belief system, which shows the presence of spiritual knowledge and of decision making. Although the individual's beliefs are considered guidelines, decision making is a complex process influenced by all forms of knowledge (i.e., rational, emotional, and spiritual) and their dynamics [16,50,51].

Maxwell [52] posited that wisdom includes knowledge and understanding, although it goes further than that by including "the desire and active striving for what is of value, the ability to see what is of value, actually and potentially, in the circumstances of life, the ability to experience value, the capacity to help realize what is of value for oneself and others" (p. 79). The key recurring word in this explanation is "value", which leads to the idea that understanding wisdom means understanding the framework of values of any organizational context where managers make decisions. Therefore, the transition from knowledge to wisdom is conditioned by the capacity to realize the potential and active values associated with certain decision making.

Hall [53] remarked on another important aspect of wisdom—the capacity to deal with uncertainty and ambiguity due to the absence of knowledge. "Wisdom is based upon knowledge, but part of the physics of wisdom is shaped by uncertainty. Action is important, but so is judicious inaction. Emotion is central to wisdom, yet emotional detachment is indispensable. A wise act in one context may be sheer folly in another" [53] (p. 11). Once again, the theory of knowledge fields, with its rational, emotional, and spiritual knowledge, is essential in understanding the complexity of the wisdom concept. Moreover, the capacity of people to deal with uncertainty makes the difference between their levels of wisdom. Uncertainty is a characteristic of the future. Therefore, wisdom is related to the capacity of people to think strategically when making decisions. In fact, evaluating the wisdom of a certain person is something we can do after a decision is made and we see the consequences. This implies time and vision for a possible future. For example, upon learning about the knowledge loss risk from an organization, those who think strategically develop solutions for knowledge retention and mitigation of the risk consequences. Intergenerational learning is one of the best solutions when there are organizational structures with generation layers [54,55].

Nonaka and Toyama [56] conceived strategic management as a distributed practical wisdom (phronesis). "Phronesis is the ability to judge goodness for the common good. This kind of judgment requires a higher point of view to be able to see what is good for the whole, even though that view stems from one individual's values and desires" [56] (p. 380). Thinking about the DIKW hierarchy and the transition from knowledge to wisdom, it is obvious that a key role is played by the spiritual knowledge field and the capacity of leaders to understand and apply in their decisions the values of goodness and the common good, and to create a vision for the company aligned with those values [44].

## 3. Materials and Methods

In the first part of this paper, we presented a critical literature review of the most significant papers and books focusing on the DIKW hierarchy. The review was based on metaphorical thinking and semantic analysis [9,25,42,43].

In the second part of this paper, we present a bibliometric analysis of the literature, focusing on the semantic clusters generated by the concept of wisdom in the literature using the specialized software VOSviewer version 1.6.16 [57–59]. The goal of a bibliometric analysis is to understand the patterns and trends in scientific research, such as the productivity of researchers, the impact of their work, and the relationship between different fields of research. A bibliometric analysis is typically used in several different ways: citation analysis—which examines the numbers and patterns of citations of a particular paper, author, or field of research to measure the impact and influence of the work; co-citation analysis—which identifies the most frequently co-cited papers, authors or journals in order to reveal the key works and researchers in a field; co-occurrence analysis—which points out the most frequently co-occurring words in titles or abstracts of articles to identify the key concepts and themes in a field; and impact analysis—which calculates indicators using citation counts, the h-index, or the journal impact factor. Many of these analyses have numeric, textual, and visual representations [60,61]. We used the default parameters. We are aware of the different types of analyses and indicators, but we considered relevant for the present analysis only the cluster co-occurrence analysis. The cluster resolution was enlarged and the minimum number of items for clusters is 1.

For this bibliometric analysis, we used the SCOPUS database and obtained 5092 results when searching for "wisdom" AND "management" as keywords. The data were downloaded on 6 January 2023. When generating the keyword co-occurrence analysis, we considered a minimum number of occurrences equal to 15. Out of the total 10,605 keywords, only 50 met the threshold. Increasing the threshold from 3 to 15 implies also increasing the significance rate of the revealed keywords and cluster composition. This is a conceptual paper based on metaphorical thinking and semantic analysis, and any nuances or any special forms of words could be relevant within the whole picture. Each of the identified words has an important meaning inside the cluster, and their impact on the subject is explained in the Results and Discussion section. For example, phronesis, practical wisdom and spirituality are not synonyms; they are distinct words with particular meanings, but they can be used with a certain degree of approximation. The query string for our analysis can be illustrated as follows:

(TITLE-ABS-KEY ("wisdom") AND TITLE-ABS-KEY ("management")).

In order to clarify all the methodological steps followed by the authors, including the criteria adopted to select the documents (e.g., documents considered, language, time windows considered), we elaborated a diagram that represents the workflow of the bibliometric analysis (Figure 2). The VOSviewer bibliometric analysis flowchart includes several sections that help to organize the process of data analysis and visualization [62].

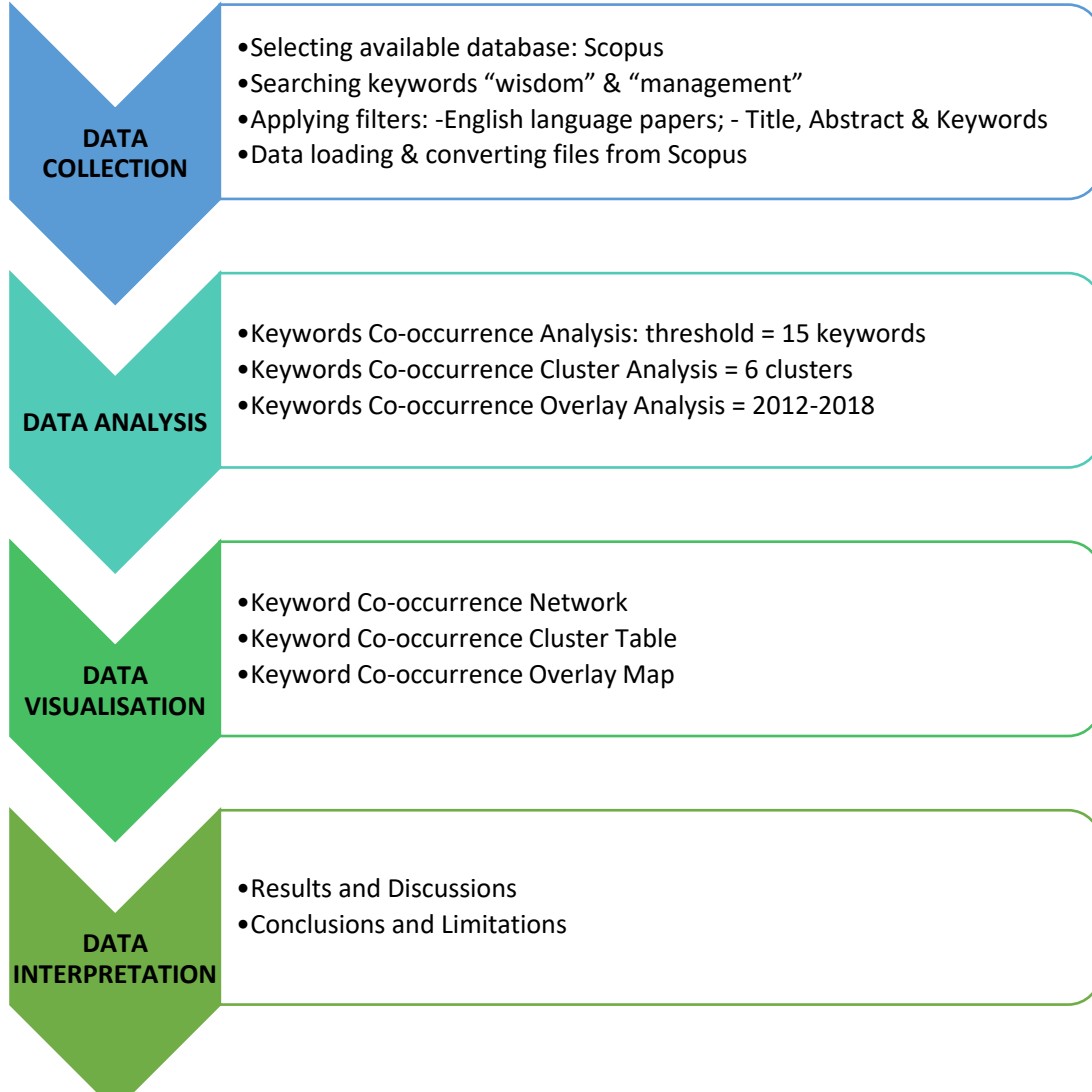

**Figure 2.** Workflow of the bibliometric analysis. (Source: Authors' elaboration).

## 4. Results and Discussion

Regarding the analyzed time period, we did not want to restrict this analysis; rather, we were interested in seeing a complete and updated evolution of the topic. Thus, the analyzed duration is the entire period for which there are publications including the indicated keywords. From the graphic, we can see that the analyzed period starts in 1943 (the year in which the first paper was published) and continues up to the present, 2023. According to Figure 3, the publications on the theme of "wisdom management" evolved significantly starting from the 2000s, then in the period 2006–2013 increased slightly and in the years 2014–2019 had an oscillatory evolution. An upward trend was recorded in the years 2017 and 2018, after which the years 2019 and 2020 showed a smaller number of publications. It will be interesting to analyze the next period and the connections that will develop with other fields, such as data science and artificial intelligence.

Performing the standard VOSviewer [63] analysis, we obtained six clusters, as shown in Figure 4 and detailed in Table 1, containing only the most relevant terms related to the investigated topic. The dataset was accurately screened for duplicates with the same meaning and form. It can be observed from Figure 4 and Table 1 that the same words do not appear twice. For our research, the keyword co-occurrence analysis was the most

relevant, and we chose to present it in detail, both as the map in Figure 4 and as clusters with values in Table 1.

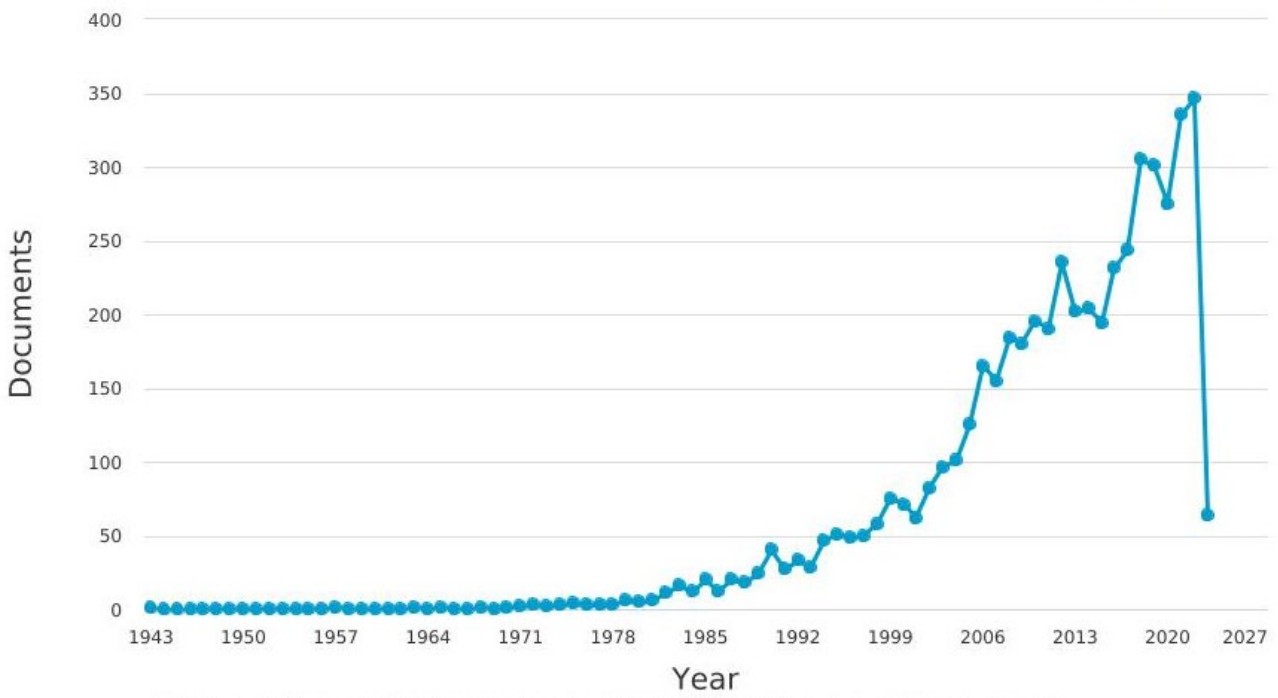

**Figure 3.** Documents by year. (Source: Extracted from Scopus).

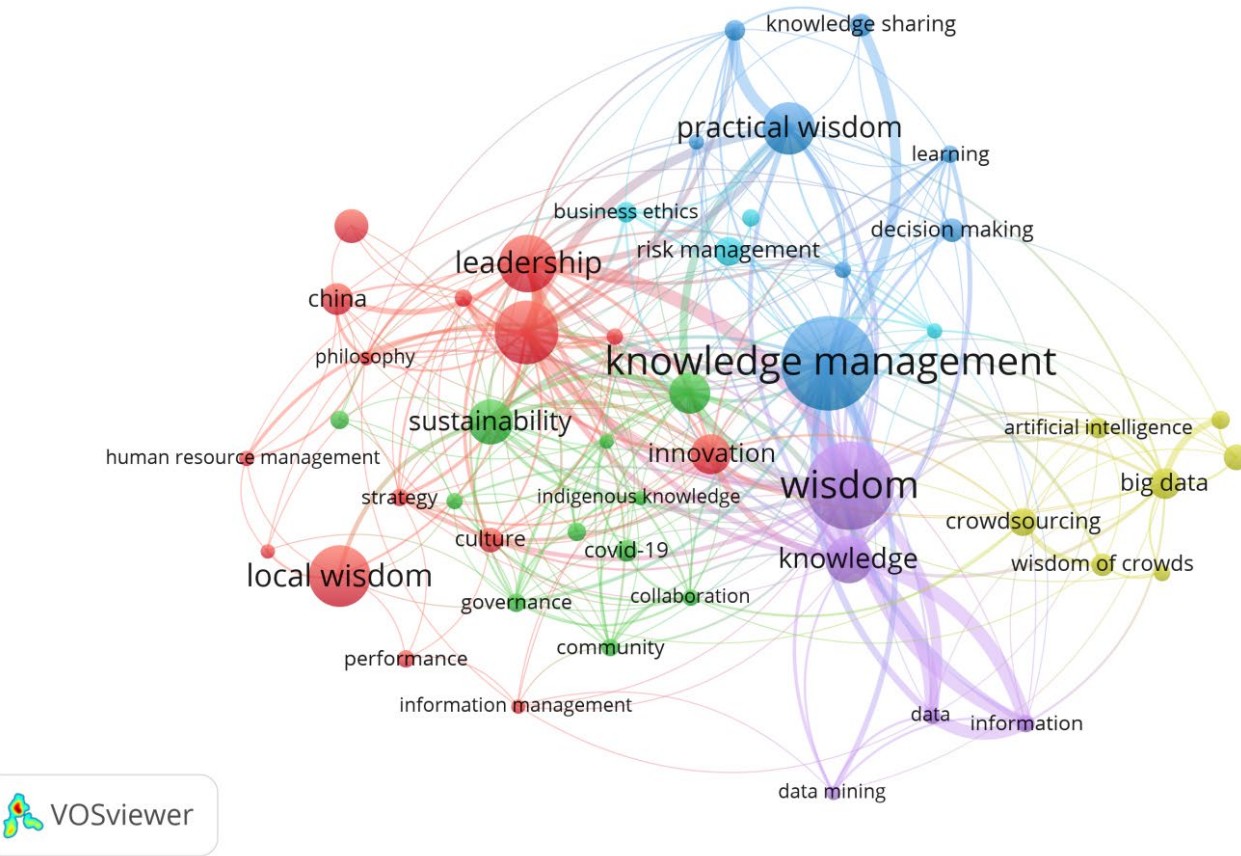

**Figure 4.** Keyword co-occurrence network visualization. (Source: Authors' results based on VOSviewer).

**Table 1.** Composition of keyword clusters by total link strength (TLS) and co-occurrences.

| Cluster Name | Items | Total Link Strength (TLS) | Co-Occurrences |
|---|---|---|---|
| MANAGEMENT AND LEADERSHIP [red cluster] | China | 18 | 35 |
| | Culture | 21 | 27 |
| | Development | 16 | 17 |
| | HRM | 9 | 16 |
| | Indonesia | 6 | 15 |
| | Information management | 9 | 15 |
| | Innovation | 37 | 46 |
| | Leadership | 83 | 69 |
| | Local wisdom | 12 | 74 |
| | Management | 84 | 78 |
| | Performance | 4 | 18 |
| | Philosophy | 21 | 15 |
| | Strategy | 23 | 18 |
| | Supply chain management | 4 | 38 |
| | Technology | 19 | 18 |
| ETHICS AND SUSTAINABILITY [green cluster] | Climate change | 14 | 20 |
| | Collaboration | 21 | 17 |
| | Community | 19 | 18 |
| | COVID-19 | 9 | 23 |
| | Ethics | 60 | 48 |
| | Governance | 25 | 20 |
| | Indigenous knowledge | 13 | 15 |
| | Policy | 12 | 17 |
| | Resilience | 10 | 15 |
| | Sustainability | 47 | 53 |
| | Sustainable development | 6 | 20 |
| KNOWLEDGE MANAGEMENT AND PHRONESIS [blue cluster] | Decision making | 17 | 26 |
| | Education | 15 | 17 |
| | KM | 114 | 121 |
| | Knowledge sharing | 14 | 24 |
| | Learning | 21 | 18 |
| | Phronesis | 30 | 22 |
| | Practical wisdom | 59 | 62 |
| | Spirituality | 11 | 16 |
| BIG DATA AND CROWDSOURCING [yellow cluster] | Artificial intelligence | 17 | 22 |
| | Big data | 31 | 34 |
| | Crowdsourcing | 17 | 31 |
| | IoT | 8 | 28 |
| | Smart city | 10 | 19 |
| | Social media | 5 | 17 |
| | Wisdom of crowds | 12 | 24 |

**Table 1.** *Cont.*

| Cluster Name | Items | Total Link Strength (TLS) | Co-Occurrences |
|---|---|---|---|
| DIKW [purple cluster] | Data | 46 | 17 |
| | Data mining | 16 | 15 |
| | Information | 47 | 18 |
| | Knowledge | 122 | 55 |
| | Wisdom | 151 | 112 |
| BUSINESS ETHICS AND DECISION MAKING [turquoise cluster] | Business ethics | 22 | 22 |
| | Decision making | 20 | 16 |
| | Project management | 2 | 18 |
| | Risk management | 7 | 32 |

The keyword network (Figure 4) shows clusters differentiated by colors that highlight groups of related entities. The clustering technique groups the keywords from the database according to the frequency of their common occurrence (co-occurrence) in different papers. Within a cluster, the links between terms are much stronger than those with terms from other neighboring clusters in the network. The colors of the clusters are predefined and intended to facilitate the visualization of patterns and relationships in the dataset. For the interpretation of the network's clusters, we must know that the terms placed centrally and with large dimensions are the most significant by comparison with those terms placed toward the network [59,64]. The fact that certain terms have a low frequency of occurrence does not mean that they are not important but rather that they are not as often encountered for various objective reasons, such as they belong to a related field, they belong to a research topic that has been overtaken by the evolution of research, or they belong to a very recent topic which recorded a small number of publications. In what follows, we will analyze in detail the clusters, their composition, and their significance.

*4.1. The Red Cluster (1): Management and Leadership*

First of all, we would like to note the visual construction of this keyword co-occurrence network. From the first view, we can easily see that the six clusters are compact and homogeneous, that is, the terms that make up each cluster are very close to each other and connected with many links, which means a high frequency with which they are used together [57].

The red cluster is the largest and is ranked by the software as the main one. This cluster integrates the key terms most frequently used together and which gravitate closest to each other as the thematic niche (see Table 1). The methodology approach of naming the clusters consists of using the term with the highest values, although in this situation we have chosen a coupling option of two of the relevant terms, namely Management and Leadership. The two items have the highest values in the cluster for both indicators, namely total link strength (TLS) and co-occurrences, as can be seen in Table 1. The red cluster highlights the fact that the basic concepts in the field of Management and Leadership, such as Innovation, Strategy and Culture, remain in the foreground in most publications. These fundamental concepts are associated with the notion of "local wisdom" in the context of research focused on "wisdom management", which makes the connection between traditional culture, community wisdom and modern management theory [65,66]. Practicing local wisdom in management and leadership is similar to practical wisdom or phronesis, and it is related to the shared values of the community. By considering local wisdom, managers can gain a deeper understanding of the cultural and social context in which their organization operates and develop more effective strategies for achieving their goals. Additionally, incorporating local wisdom can help to foster a sense of community and shared ownership among

employees and stakeholders, which can lead to increased motivation and commitment to the organization's mission [17,33,44,53].

### 4.2. The Green Cluster (2): Ethics and Sustainability

We chose the title in accordance with the two most significant terms because the values are extremely close and, on the network, the two positions coincide and almost entirely overlap: "sustainability" has a TLS = 47 and a number of occurrences = 53, and "ethics" has a TLS = 60 and a number of occurrences = 48. Analyzing the composition of the green cluster, the two terms Ethics and Sustainability become emblematic of research contexts such as Climate change, Governance, Indigenous knowledge, Policy, Resilience, and Sustainable development. In this cluster, a key phrase appears again that connects us to the basic concept of "wisdom management", namely "indigenous knowledge". From the map, we can see that this expression has a central position, which indicates very good representativeness in the discussed thematic area. From this context, as well as from the literature, we deduce that the expression "indigenous knowledge" is related to "local wisdom", signifying the knowledge and practices specific to a certain culture or community, which have evolved over time and are considered relevant for solving problems which are specific to that community.

### 4.3. The Blue Cluster (3): Knowledge Management and Phronesis

This cluster is extremely relevant for the purpose of the present research. For the name of this cluster, we have a combination of two concepts, namely "knowledge management" and "phronesis" or "practical wisdom". The last two forms have equivalent meanings, although their use depends on the authors' contextual perspectives [32,33]. We can observe in the literature an increasing frequency of using these concepts, especially after Nonaka and Takeuchi published their seminal book on the knowledge-creating company, where they argued for the importance of the spiritual dimension of knowledge management [10]. In addition, Nonaka and Zhou underlined the key role played by the organizational context in the process of decision making [17]. Both the concepts of phronesis and practical wisdom integrate knowledge with the act of decision within a certain context. This can be seen from Figure 4 when looking for the connections between "practical wisdom" and "ethics" and "decision making".

The concept of "knowledge management" has spread to many fields of specialized literature, although it is still a current topic for which notions such as Decision making, Education, Knowledge sharing, Learning, Phronesis, Practical wisdom, and Spirituality are intensively researched. The in-depth analysis of the concepts integrated by this cluster leads us to a series of perspectives on spirituality in knowledge management. Spirituality can help us to look at knowledge management from the perspective of the whole instead of focusing only on functional or instrumental aspects. It can provide a moral and value framework for knowledge management, helping to establish long-term goals and objectives. Spirituality can stimulate creativity and innovation by connecting with inner sources of wisdom and inspiration. Regarding communication and collaboration, spirituality can contribute to deeper and more authentic processes between team members and partners. Spirituality may also contribute to developing an attitude of responsibility and sustainability in knowledge management by connecting with social and ecological values and goals [67]. This cluster shows the links between the knowledge and wisdom levels of the DIKW hierarchy, and its connection with the other clusters demonstrates how authors have enlarged the horizon of wisdom beyond that of the hierarchy. Figure 4 shows us the presence of the "COVID-19" term, which illustrates how researchers studied the impact of the global crisis generated by the COVID-19 pandemic on knowledge management. Practically, the crisis revealed how difficult it was for many companies to find the best strategies in the absence of knowledge concerning the new virus and the unexpected global measures against the pandemic. Developing knowledge strategies could be useful in preventing such crises or in reducing their severe consequences.

### 4.4. The Yellow Cluster (4): Big Data and Crowdsourcing

The yellow cluster is strongly highlighted on the right side of Figure 4, although it has a significantly smaller number of terms: Artificial intelligence, IoT, Smart city, Social media, and Wisdom of crowds. Although it is focused on information technology, there is a direct connection with "wisdom management", namely "wisdom of crowds", which is directly linked with "crowdsourcing" (i.e., knowledge sources of crowds). The analysis of this cluster shows, in an obvious way, the connection between theory and practice, between conventional and digital practices [68]. Big Data and Crowdsourcing provide the content for the lower levels of the DIKW hierarchy and contribute to their upward transitions.

### 4.5. The Purple Cluster (5): The DIKW Hierarchy

This cluster contains all the links between publications dedicated to data, information, knowledge, and wisdom and their hierarchy. The high LTS numbers for knowledge and wisdom (i.e., 122 and 151), as well as their numbers of co-occurrences (i.e., 55 and 112), show the good connection between these concepts and the hierarchy dynamics between them. Figure 4 shows good connections with the red and the blue clusters, which means knowledge management, leadership, and phronesis, supporting the critical literature analysis we performed in the previous section of the present paper.

The DIKW Hierarchy cluster is important for knowledge management when it comes to understanding the transitions from data to information, from information to knowledge, and from knowledge to wisdom, as well as their role in contributing to the organizational knowledge capital and its dynamics [69,70]. Our analysis demonstrates that the links between the published papers support the DIKW hierarchy, although the transition from knowledge to wisdom is long and nonlinear. It encompasses a larger area than that of the DIKW hierarchy.

### 4.6. The Turquoise Cluster (6): Business Ethics and Decision Making

The last but not the least cluster, the turquoise color, we named after the same pattern, including two of the representative terms. The composition of this cluster, as shown in Table 1, highlights the close links between the decision-making process, ethics and wisdom management. By connecting the wisdom of the crowd in the digital environment with knowledge management, a broader and more diverse perspective on the problem can be obtained and better decisions can be made. In this context of discussions, the topic of business ethics could not be missing because it has become an essential part of this complex and extremely dynamic ensemble. Ethics in the management process and practical wisdom are strongly connected through the set of organizational values. A synthetic analysis of the clusters identified by VOSviewer is presented in Table 1.

### 4.7. The Overlay Map Analysis

The overlay map in the bibliometric analysis is most relevant for the visualization of the relationships between different concepts or keywords within a specific field of research or study. Overlay maps can be used to identify key contributors and influential publications within a particular field across a certain period of time. The particularity of this representation is that it shows in gradient colors the yearly evolution of research in the field. In fact, associating the keywords' colors with the years' colors on the graphic legend facilitates the understanding of the subject's chronological evolution within the academic literature. This can be useful for researchers and practitioners looking to stay current in their field or for those looking to build on existing research [17,58,59]. An overlay map can be a valuable tool for gaining a better understanding of a specific area of study and for identifying areas of potential growth and opportunity. Figure 5 presents an overlay map of our inquiry concerning wisdom and its semantic links with knowledge and knowledge management. The map shows how the discussions evolved from "knowledge" and "knowledge management" in 2012, to "sustainability" in 2014, to "crowdsourcing" in 2016, and recently, to "local wisdom" and "phronesis" in 2018. Year by year, the focus of

researchers has shifted to related concepts, and each year provides a different combination of concepts that generate messages for different theory or practice issues.

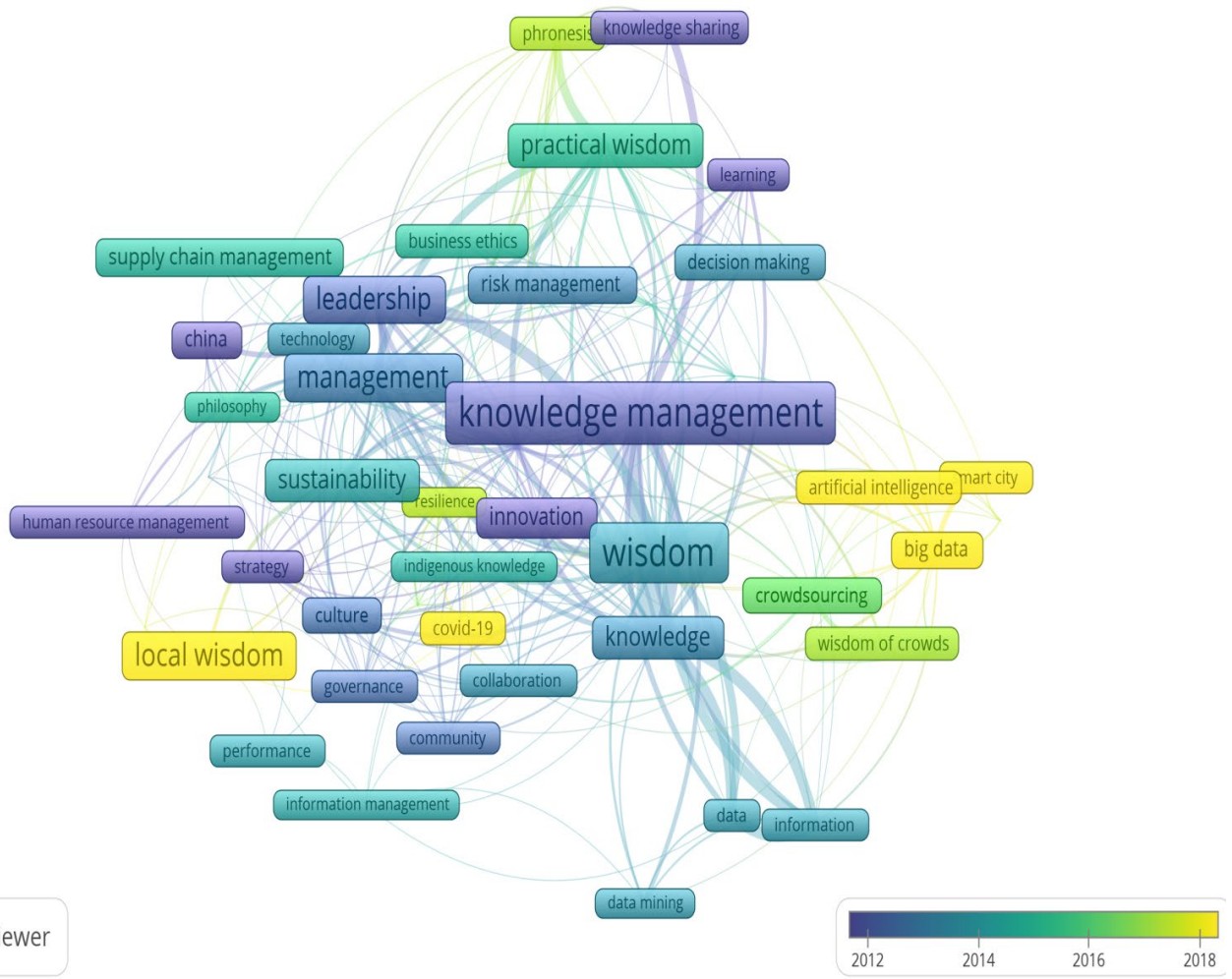

**Figure 5.** Overlay map 2012–2018. (Source: Authors' results based on VOSviewer).

It is very interesting to observe the extremes of the chronology map. The basic idea that we deduce is that in 2012, the specialized literature discussed knowledge management (dark-blue cluster) in the classic sense of organizational implementation, while in 2018, the debates on the same theoretical concept of knowledge management were strongly connected with the field of technology, as shown by the yellow cluster on the overlay map. Additionally, wisdom's connection with knowledge management is evolving over time, with more focus on wisdom and practical wisdom.

## 5. Conclusions

The present paper aims at analyzing critically how the DIKW hierarchy is reflected in the literature and how its assumptions remain valid in the new paradigm of understanding the concepts of knowledge and wisdom as well as the transition from knowledge to wisdom. In the beginning of developing knowledge management systems, the DIKW hierarchy appeared as a useful tool for understanding the structure of the fundamental intangible resources (i.e., data, information, knowledge, and wisdom). Moreover, it was a simple and intuitive explanation of the transition from data to information, from information to knowledge, and from knowledge to wisdom. Today, the DIKW hierarchy may represent a first step in explaining the connections between data, information, knowledge, and wisdom, although we need to go beyond the framework created by this hierarchy and to use the new models for explaining knowledge and wisdom as intangible resources for an organization.

Special attention should be paid to the distinction between the interpretations of the concept of "information" in the fields of information systems management, linguistics, philosophy, and knowledge management systems. Shannonian information is a concept devoid of any meaning, being a mathematical abstraction. The same is true of information entropy. However, semantic information represents an extension of the mathematical concept to linguistics and philosophy and has new features. Finally, the concept of information used in knowledge management represents data with meaning, which is totally different with respect to the mathematical definition given by Shannon.

For knowledge, things are more complicated because its definition in management depends on the metaphor used and not on the simple transition from data. Knowledge flows and knowledge fields constitute new approaches that reveal the complexity of the semantic domain of knowledge and its dynamics. It remains based on information but in a more sophisticated way. The theory of knowledge fields brings forth the three basic types of knowledge (i.e., rational, emotional, and spiritual), which have different influences on decision making and actions in management. From the DIKW hierarchy perspective, spiritual knowledge becomes more important because it dominantly influences the transition from the level of knowledge to the level of wisdom in the hierarchy.

Wisdom represents the pinnacle of the DIKW hierarchy, and it is the most complex concept applied in management. Wisdom becomes phronesis or practical wisdom in management and is critical to decision making. Wisdom contains knowledge, especially spiritual knowledge, but it also includes the decision act. When managers and leaders practice wisdom in a systematic way, we may discuss a wise company that is capable of achieving a sustainable competitive advantage.

The bibliometric analysis with VOSviewer reveals new features of the semantic links between all of these concepts and how they offer a larger perspective on the position of wisdom with respect to knowledge management than we get from the DIKW hierarchy. Our analysis shows six main clusters: (1) The red cluster—Management and Leadership; (2) The green cluster—Ethics and Sustainability; (3) The blue cluster—Knowledge Management and Phronesis; (4) The yellow cluster—Big Data and Crowdsourcing; (5) The purple cluster—The DIKW Hierarchy; and (6) The turquoise cluster—Business Ethics and Decision Making. Analyzing each cluster generated by VOSviewer, we get a deeper understanding of how the literature reflects dynamically the research focused on the long distance from knowledge to wisdom in management. We remarked especially on the Blue cluster revealing the links between knowledge management, phronesis and practical wisdom, and on the Purple cluster focusing on the major concepts of the DIKW hierarchy, data, information, knowledge, and wisdom. Here, we remark on the high TLS values for the concepts of knowledge (122) and wisdom (155), suggesting the importance in that hierarchy and their dynamics.

The limitations of our research come first from the fact that we worked only with the Scopus database. It is necessary in a further analysis to work with Web of Science and other significant databases from the topic point of view. Another limitation refers to the parameter selection. Ignoring keyword similarities, plurals, non-standard spellings or acronyms might distort the conclusions. In addition, we focused more on the first part of the paper than on the bibliometric analysis, because we found in the literature gaps and ambiguities in interpreting the basic concepts of the DIKW hierarchy. A more detailed bibliometric analysis could reveal new aspects that may be of interest for the knowledge management domain.

**Author Contributions:** The following work has been categorized under different authors' names. Conceptualization: C.B. and R.B.; Methodology: C.B. and R.B.; Validation: C.B. and R.B.; Formal analysis: C.B. and R.B.; Investigation: C.B. and R.B.; Data curation: C.B. and R.B.; writing—original draft preparation: C.B. and R.B.; Writing—review and editing: C.B. and R.B.; Visualization: C.B. and R.B.; Supervision: C.B. and R.B.; Project administration: C.B.; Funding acquisition: N/A. All authors have read and agreed to the published version of the manuscript.

**Funding:** This research received no external funding.

**Institutional Review Board Statement:** Not applicable.

**Informed Consent Statement:** Not applicable.

**Data Availability Statement:** Not applicable.

**Conflicts of Interest:** The authors declare no conflict of interest.

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
