# Peer review of "From Knowledge to Wisdom: Looking beyond the Knowledge Hierarchy"

_knowledge, doi:10.3390/knowledge3020014_

Round 1

Reviewer 1 Report

The paper discusses a valid topic and presents a novel literature review. It is well-written and structured. The content is logic and does not have any flaws. I have no suggestions for improvement. Congratulations on the paper!

Author Response

Please see the attached pdf file.

Reviewer 2 Report

This is a timely contribution to the field of wisdom and management with the potential to translate to other fields, namely psychology and education. The structure is clear and coherent. One revision I strongly suggest regarding methods is clarifying the strategy to manipulate the Scopus data, for example, were the data screened for duplicates?
Are all documents articles, were there any strategies to prioritize peer-reviewed research vs chapters or nonpeer-reviewed documents? 
While VOS display allows to build co-citation networks, it only focuses on the likelihood of two keywords being used together. Other analyses, like bibliometric coupling or co-occurrence networks, can add nuance by simultaneously considering the impact (number of citations) of the documents and the keywords they contain. 

50 keywords with at least 15 co-occurrences in 10k + seems a low number of co-occurrences. Please provide a table with relative and cumulative frequencies of these top keywords in relation to the full keyword dataset.

Could you please provide an explanation of how keywords were treated? For example, how are synonymous keywords counted? Often times bibliometric software overlooks synonyms, spellings, and ambiguous words, unnecessary words, and poorly standardized words across journals and authors. For example, the word China is listed as a high-frequency word. However, what does it mean in the context of the study? Authors may tag their countries as keywords, however, the keyword may not be part of the central theme of the research. 
Finally, grouping and replacing keywords may provide a better landscape of how words are used, preventing noise in the analysis by non-standardized keywords, and amplifying the current results by covering higher frequencies in word counts. Likey, these methodological issues could shed different results, therefore, results need careful revision as well.

Author Response

Please see the attached pdf file.

Reviewer 3 Report

From Knowledge to Wisdom: Looking Beyond the Knowledge Hierarchy

by Constantin Bratianu, Ruxandra Bejinaru

Overview

The manuscript focus on the transition from knowledge to wisdom, looking beyond the classical wisdom hierarchy. Furthermore, the authors analyzed the connection between knowledge and wisdom by using a bibliometric approach.  In this perspective, the authors made use of the Scopus database and the VOSViewer software.

The subject certainly deserves a scientific study and indeed the work is quite interesting. However, the manuscript needs substantial revisions.

Firstly, the general structure of the manuscript does not show the classic subdivision into Sections: Introduction, Materials and Methods, Results, Discussion (or "Results and Discussion"), and Conclusions. This makes difficult to follow the logical thread of the work. Therefore, the manuscript must be arranged following the structure indicated in the "Instruction for Authors" section of the Journal.

Other suggestions aimed to improve the manuscript are reported below.

Point by-point analysis

Abstract

At the end of the section, the authors will include a paragraph explaining who and why can benefit of the study results.

3. A bibliometric analysis of the wisdom's semantic links

This section will be named "Materials and Methods", where all the criteria used will be detailed. At present, the discussion of the data and the methodology used are insufficient and confusing. Indeed, the bibliometric study requires asystematic protocol that allows the reaserch to be reproducible and reliable. This is essential to make the research sound.

Please add a flow chart able to clarify the whole path followed for the research, including the criteria used to select the documents (e.g. typology of documents, language, time span considered, format in which the data was downloaded to be processed by VOSViewer).

Line 358. The authors will separate the citations of bibliographic references relating to the software from those relating to its use by the scientific community. In that perspective, references 13 and 14 quoted in line 65 (Introduction) will be separated and quoted here.

Lines 358-360. The statement requires 1-2 references.

Line 369. The results must be discussed in the dedicated section (Results) which should include a graph with the number of documents over time.

Line 374-376. As indicated above, the results must be reported in the dedicated section (Results).

The authors used Scopus for the bibliographic data search. However, it is required to justify the choice. Why didn't the authors consider other databases (e.g. Web of Sciences)? Was a comparison made between different databases? Which deductions can be made after the comparison? Furthermore, the main features of Scopus citation index need be discussed.

The authors will clarify in which field ("Article Title" or "Article title, Abstract, Keywords") of Scopus the keywords "wisdom" and "management" were searched.

When was the data downloaded?

The introduction of the software VOSViewer will be made after having discussed the source of data (Scopus). Furthermore, the authors should indicate the software version used.

The authors must include in the methodological section the discussion regarding both the different typologies of maps (network visualization, overlayvisualization, density visualization)  that can be obtained by the software and the meaning of TLS (Total Link Strenght). Furthermore, the authors must indicate all the software parameters used (e.g. cluster resolution, minimum number of items for clusters). Specify if default parameters were used.

The Overlay map analysis

The years shown in the overlay map are average values. This should be explained. Also, the overlay map explanation is quite poor. For example, the authors could discuss the recent use of the term "phronesis" versus "practical wisdom", having the discussion related to the authors' contextual perspective (lines 443-444) in mind. Why did the term emerge in more recent times? Which explanation can the authors provide? Furthermore, in the map, I also read "covid-19" and "resilience" terms. Considering these terms, what can be argued with respect to the "knowledge" and "knowledge management"?

 Conclusions

The authors will have to better summarize the results of the bibliometric study, clearly highlighting the conclusions they have reached by this approach, linking the arguments to the research question posed in the Introduction. Suggestions for stakeholders and practitioners will also be particularly useful.

Finally, the authors will discuss the limits of their research, highlighting the possible research scenarios towards which the scientific community should address.

Author Response

Please see the attached pdf file.

Round 2

Reviewer 2 Report

Thanks for your response in adding details to the methods sections. I understand that the bibliometric analysis is secondary to the manuscript's purpose. However, I suggest some of the methodological weaknesses may be more amply discussed in the limitations section. 

For example, when keywords are not systematically treated, it is possible to attenuate seemingly extensive relationships among words or even exacerbate the strength of identified links. Therefore, ignoring keyword similarities, plurals, non-standard spellings or acronyms might distort the conclusions. 

Author Response

Thank you for your observations and recommendations.

Comments and Suggestions for Authors:

Thanks for your response in adding details to the methods sections. I understand that the bibliometric analysis is secondary to the manuscript's purpose. However, I suggest some of the methodological weaknesses may be more amply discussed in the limitations section. 

For example, when keywords are not systematically treated, it is possible to attenuate seemingly extensive relationships among words or even exacerbate the strength of identified links. Therefore, ignoring keyword similarities, plurals, non-standard spellings or acronyms might distort the conclusions. 

Answer:

We agree with your recommendation and inserted in the last section of the paper the following text:

“Another limitation refers to parameters’ selection. Ignoring keywords similarities, plurals, non-standard spellings or acronyms might distort the conclusions.”

Reviewer 3 Report

The authors performed many improvements. However, two main points still need to be addressed by the authors.

1) I do not see the graph showing the number of documents over time. As already requested in the first review, it is necessary to clarify which time frame the 5092 documents found in Scopus cover and how many documents per year were published. The graph will be discussed.

2) Line 398-399. This is not a flowchart I required in the first review, but the query string. I required the flowchart (the diagram that represents the workflow) to clarify all the methodological steps followed by the authors, including the inclusion and exclusion criteria (e.g. documents considered, language, time windows considered) adopted to select the documents.

Line 412. Delete the references [57-59] already included in the line 372. These articles can be replaced by others highlighting the consolidated use of the software in different scientific areas.

Author Response

Thank you for your observations and recommendations.

Reviewer #3 (Round 2)

Comments and Suggestions for Authors

The authors performed many improvements. However, two main points still need to be addressed by the authors.

Comment #1.

 I do not see the graph showing the number of documents over time. As already requested in the first review, it is necessary to clarify which time frame the 5092 documents found in Scopus cover and how many documents per year were published. The graph will be discussed.

Answer #1.

We inserted in the paper the following text and figure:

“Regarding the analyzed time period, we didn’t want to restrict this analysis, but rather we were interested in seeing a complete and updated evolution of the topic. Thus, the analyzed duration is the entire period for which there are publications including the indicated keywords. From the graphic we can see that the analyzed period is starting 1943 (the year in which the first paper was published) and up to the present, 2023. According to the graphic, the publications on this theme of "wisdom management" have evolved significantly starting with the 2000s, then in the period of 2006 -2013 increased slightly and in the years 2014-2019 they had an oscillatory evolution. An upward trend was recorded in the years 2017 and 2018, after which the years 2019 and 2020 show a smaller number of publications. It is interesting to analyze the next period and the connections that will develop with other fields such as data science and artificial intelligence.

Comment #2.

Line 398-399. This is not a flowchart I required in the first review, but the query string. I required the flowchart (the diagram that represents the workflow) to clarify all the methodological steps followed by the authors, including the inclusion and exclusion criteria (e.g. documents considered, language, time windows considered) adopted to select the documents.

Answer #2.

We introduce in the paper the following Flowchart

Figure 2. Methodological FLOWCHART

Comment #3.

Line 412. Delete the references [57-59] already included in the line 372. These articles can be replaced by others highlighting the consolidated use of the software in different scientific areas.

Answer #3.

We deleted the references [57-59] as requested. We introduced a new reference [63] (due to introducing two new reference, their numbers changed).
